# Atmospheric Pressure Cold Plasma Modification of Basil Seed Gum for Fabrication of Edible Film Incorporated with Nanophytosomes of Vitamin D_3_ and Tannic Acid

**DOI:** 10.3390/foods12010071

**Published:** 2022-12-23

**Authors:** Hadi Hashemi Gahruie, Mohammad Hadi Eskandari, Rohollah Sadeghi, Seyed Mohammad Hashem Hosseini

**Affiliations:** 1Department of Food Science and Technology, School of Agriculture, Shiraz University, Shiraz 7144165186, Iran; 2Department of Animal, Veterinary and Food Sciences, University of Idaho, Moscow, ID 83844, USA

**Keywords:** basil seed gum, cold plasma, rheological properties, edible film, nanophytosome, encapsulation

## Abstract

The purpose of this work was to first investigate the impact of cold plasma (CP) treatment, performed at various times (0–30 min), on the characteristics of basil seed gum (BSG), as well as the fabrication of functional edible films with the modified BSG. FT-IR spectra of CP-treated BSG revealed change at 1596 and 1718 cm^−1^, indicating the formation of carbonyl groups. Both untreated and CP-modified BSG dispersions showed shear-thinning behavior with a higher apparent viscosity for the CP-modified dispersions at studied temperatures. Untreated BSG dispersion and the one treated by CP for 10 min revealed time-independent behavior, while those treated for 20 and 30 min showed a rheopectic behavior. CP-modified BSG dispersion had higher G′, G″, and complex viscosity than untreated BSG. Higher contact angle for the CP-modified BSG suggested enhanced hydrophobic nature, while the surface tension was lower compared to the untreated BSG. SEM micrographs revealed an increase in the surface roughness of treated samples. Moreover, modified BSG was successfully used for the preparation of edible film incorporating tannic acid and vitamin D_3_-loaded nanophytosomes with high stability during storage compared to the free form addition. The stability of encapsulated forms of vitamin D_3_ and tannic acid was 39.77% and 38.91%, more than that of free forms, respectively. In conclusion, CP is an appropriate technique for modifying the properties of BSG and fabrication of functional edible films.

## 1. Introduction

Basil (*Ocimum basilicum* L.), from the Lamiaceae family, is a native herb of Africa, Asia, and South and Central America. Due to the high dietary fiber content, as well as its therapeutic and rheological properties, basil seed and basil seed gum (BSG) are utilized as an ingredient in some types of beverages and desserts [1]. The major monomeric constituents of BSG are D-glucose, D-galactose, D-mannose, L-rhamnose, and L-arabinose [2]. Different properties of BSG, such as biocompatibility, hydrophilicity, edibility, economic cost, viscoelastic nature, and its ability to form edible films, make BSG a good candidate in food applications [3,4].

The emulsifying ability of BSG at different conditions has been investigated in several studies [2]. Osano et al. [5] studied the emulsifying ability of BSG at different concentrations and after removing the protein portion of the gum. The effect of heating, pH, and ionic strength on the emulsifying property of BSG was also investigated by Hosseini-Parvar et al. [6]. They declared that the main reason for the physical stability of the emulsion at extreme ionic strengths and pH values was the gel-like behavior of gum in the aqueous phase of the emulsion.

Gum modification has recently attained considerable attention in the food, cosmetics, and medical industries. The chemical modification of BSG using octenyl succinic anhydride (OSA) and its effects on the rheological and interfacial [7], as well as film forming [1] properties of BSG, were previously reported. Atmospheric pressure cold plasma (CP) is a relatively novel technology, which can modify the surface and physicochemical properties of different biopolymers at low temperatures [8,9] in several ways, including surface activation, surface etching, cross-linking, deposition, and grafting [10]. The effects of CP treatment on gum Arabic [11] and xanthan gum [12,13] revealed that this technique is effective for developing advanced new materials. Limited reports are available on the characteristics of CP-treated natural gums. To the best of our knowledge, the effect of CP treatment on the physicochemical and rheological properties of BSG has not yet been reported. Therefore, BSG was subjected to CP treatment at different times and then characterized to modify its characteristics. Furthermore, edible films were made using the modified BSG, and its properties and potential to incorporate vitamin D3 and tannic acids, in the form of nanophytosomes, were studied.

## 2. Materials and Methods

### 2.1. Materials

Basil seed gum (BSG) powder was purchased from Reyhan Gum Parsian Co. (Tehran, Iran). As expressed by the supplier, the foreign bodies present in the seeds were removed in an excess amount of ethanol followed by drying at 45 °C. Dried seeds were soaked in double distilled water at 50 ± 1 °C. BSG was scraped off from the surface of basil seeds by shearing, filtered through cloth, and dried. Native BSG obtained from the same supplier was well characterized in terms of protein, carbohydrate and moisture content, molecular weight, molecular conformation, zeta potential, contact angle, rheological properties under the effect of temperature, shear rate and time, viscoelastic properties, intrinsic viscosity, surface morphology, interfacial tension, emulsifying ability, emulsion stability, foaming ability, foam stability, FT-IR and 1H NMR spectra, and crystallinity by XRD in our previously published work [1,7]. All other chemicals were of analytical grade.

### 2.2. BSG Modification with Plasma Treatment

BSG powder was treated based on the method of Kim et al. [14] at different times (10–30 min) by an in-package cold plasma source (NIK Plasma Tech., Tehran, Iran) operating at 7 kV voltage, 10 kHz frequency, and ambient air. A thin layer of copper covered the inner plastic walls of the instrument container as the surface barrier discharge (SBD). Plasma treatment between the plates of the plasma instrument was conducted for 10, 20, and 30 min. The BSG samples treated at different times were named CP_10_, CP_20_, and CP_30_.

### 2.3. Properties of Modified BSG

#### 2.3.1. Surface Morphology

Various BSG powders were adhered to stubs by an aluminum tape and coated with a gold layer (8 nm thickness). The surface morphology was then evaluated using a scanning electron microscope (SEM, TESCAN vega3, Brno–Kohoutovice, Czech Republic) at an accelerating voltage of 20.0 kV and various magnifications.

#### 2.3.2. Fourier-Transform Infrared Spectroscopy

The FT-IR spectra of control (C) and CP-treated samples were recorded in the range of 4000–400 cm^−1^ by Thermo Nicolet Avatar 370 spectrometer (Thermo Nicolet Corp., Madison, WI, USA).

#### 2.3.3. Contact Angle

Small pellets were prepared by pressing various BSG powders. A drop shape analyzer (DSA 100, KRÜSS GmbH, Hamburg, Germany) was used to determine the contact angle (θ). Briefly, the BSG pellet was placed at the bottom of a glass aquarium containing pure canola oil and then a water droplet (2 μL) was deposited on the pellet surface. Droplets photographs were taken by a CCD camera equipped with the macro lens and analyzed by the DSA software for θ measurement.

#### 2.3.4. Color Parameters

The color parameters (i.e., *L**, *a**, *b**, and ΔE) of samples were measured according to the method described by [15].

#### 2.3.5. Rheological Properties

##### Apparent Viscosity

Steady shear properties of BSG dispersions in double distilled water (0.3% *w*/*v*) were evaluated at the shear rate range of 1–301 s^−1^ at 20 °C using a rheometer (MCR 302, Anton Paar, Graz, Austria) equipped with a cone-plate geometry (type CP25-1, cone diameter = 25 mm, gap size = 0.052 mm, and cone angle = 1°). Temperature control (±0.1 °C) was carried out by a Peltier system (Viscotherm VT2, Phar Physica, Graz, Austria). Shear stress vs. shear rate data were fitted by four models including Power Law, Herschel–Bulkley, Bingham, and Casson (Equations (1)–(4), respectively).
τ = k(γ^n^)(1)
τ = τ_0_ + k(γ^n^)(2)
τ = τ_0_ + µγ(3)
τ^0.5^ = τ_0_^0.5^ + k_c_(γ^0.5^)(4)
where γ is the shear rate (s^−1^); τ is the shear stress (Pa); τ_0_ is the yield stress (Pa); k and k_c_ are the consistency coefficient (in Pa.s^n^ for Herschel–Bulkley and Power Law and Pa.s^0.5^ for Casson); n is the flow behavior index (dimensionless); and μ is the Bingham viscosity (Pa.s).

##### Temperature Sweep

The impact of heating (from 20 °C to 60 °C at a rate of 4.3 °C/min) and subsequent cooling (at a similar rate) on the viscosity of BSG dispersions (0.3% *w*/*v*) was studied at a constant shear rate of 50 s^−1^.

##### Time Dependency

Time-dependent rheological properties of various BSG dispersions (0.3% *w*/*v*) were examined at a constant shear rate of 50 s^−1^ at 20 °C. The experiment was continued until reaching a steady state.

##### Viscoelastic Properties

Dynamic rheological behavior of various BSG dispersions (0.3% *w*/*v*) was studied by the same instrument mentioned in Section Apparent Viscosity. To determine the linear viscoelastic region (LVR), amplitude sweep tests were performed in the shear strain range of 0.01–100% and a constant frequency of 1 Hz. After that, frequency sweep tests were done at 0.1% shear strain at the frequency range of 0.01–10 Hz. Before performing each experiment, the gum dispersion (2 mL) was deposited over the rheometer plate and given enough time for the structure recovery. Different rheological parameters, including storage modulus (G′), loss modulus (G″), loss factor (tan δ), and complex viscosity (η*) were obtained using RheoCompass™ software (Anton Paar, Graz, Austria).

#### 2.3.6. Surface Tension

The surface tension of control and CP-treated BGS dispersions was measured at 20 °C using a tensiometer (Nanometric, Contact angle-101, Shiraz, Iran) according to the method of Mirzapour-Kouhdasht et al. [16].

### 2.4. Nanophytosome Preparation

For the preparation of nanophytosomes, phosphatidylcholine (2 mg/mL), tannic acid (1 mg/mL), and vitamin D_3_ (1 mg/mL) were dissolved in ethanol and kept at 7 °C for 12 h, followed by evaporation in the rotary evaporator at 40 °C. This process was followed by adding double distilled water to prepare prephytosome structures. To obtain homogeneous nanophytosmes, the mixture was homogenized by a Ultra-Turrax (T25, IKA, Staufen, Germany) at 13,000 rpm for 3 min and then sonicated at 150 w for 5 min (Bandelin, Berlin, Germany) [17].

### 2.5. Film Preparation

Modified BSG (CP_30_) solution (1% *w*/*w*) was prepared in double distilled water containing glycerol (0.5% *w*/*w*) as the plasticizer and hydrated overnight under continuous mixing. Then, the nanophytosomes dispersion was mixed with the film-forming solution. Moreover, a free form of tannic acid (1000 mg/kg of the dried film) and vitamin D_3_ (1000 mg/kg of the dried film) dissolved in double distilled water was prepared as a control to investigate effectiveness of the phytosome loading strategy. The prepared dispersions (20 g) were cast onto Teflon plates (48 cm^2^) and dried at 25 °C for 72 h. Finally, dried films were peeled off and conditioned at 25 °C and 53% RH before further experiments and analyses. Different modified BSG films were termed based on the loading technique of tannic acid and vitamin D_3_ as BSG-free and BSG-phytosome. The BSG control film (BSG-C) without any added bioactive compounds was also prepared.

### 2.6. Film Characterization

#### 2.6.1. Thickness and Density

A digital micrometer was used to measure the film thickness based on the method of Mahajan et al. [18] and Ghiasi and Golmakani [19]. The film density was measured by dividing the weight of the film by its volume (surface area × thickness).

#### 2.6.2. Moisture Content and Solubility

Moisture content and solubility of films were measured by the methods modified by Moghadam et al. [20].

#### 2.6.3. Opacity and Color

Opacity was measured by dividing the absorbance values of film strips at 600 nm by their thickness. Color parameters including redness (*a**), yellowness (*b**), and lightness (*L**) were evaluated by Photoshop software CS6. The total color difference (ΔE) and whiteness index (WI) were also calculated using the equations reported by Tabatabaei et al. [21].

#### 2.6.4. Stability of Bioactive Compounds during Storage

The stability of tannic acid and vitamin D_3_ within the film matrix, either in the free form or encapsulated form in phytosome, was measured by the spectrophotometric method described by Aelenei et al. [22] and Behjati and Yazdanpanah [23], respectively.

### 2.7. Statistical Analysis

All tests were carried out three times and the results were reported as the average of replicates ± standard deviation. One-way analysis of variance at the significance level of 0.05 was used for data analysis. The means ranking was done using Duncan’s multiple range tests (SAS^®^ software, ver. 9.1, SAS Institute Inc., Cary, NC, USA).

## 3. Results and Discussion

### 3.1. Properties of Modified BSG

#### 3.1.1. BSG Microstructure

Figure 1 presents the SEM images (×20,000 magnification) of control and CP-modified BSG powders. Obviously, the surface microstructure of BSG was changed after CP treatment. The BSG samples treated by CP revealed a laminated structure and a high degree of surface roughness (due to small particles) especially at longer plasma treatment times, likely due to higher surface etching and more penetration of plasma to the gum structure. It has been reported that the CP can alter the hydrocolloids microstructure through physical (e.g., separating some parts of the gum structure) and/or chemical (e.g., degradation and chain-breaking) mechanisms [24]. Zein-based films [25] and brown rice grains [26,27] showed similar changes in morphology after treatment by plasma.

#### 3.1.2. Fourier-Transform Infrared Spectroscopy

FT-IR spectroscopy provides information regarding the functional groups of biopolymers and the changes after modification [12]. The FT-IR spectra of control and CP-treated BSG samples are shown in Figure 2. The peak at 1025 cm^−1^ was related to C-O-C stretching vibrations of glycosidic bonds. The C=N stretching vibrations appeared at 1408–1406 cm^−1^. The peak that appeared at around 1600–1596 cm^−1^ was assigned to the presence of asymmetric and symmetric stretching of carboxylate anion, which was decreased after CP treatment, particularly at a longer treatment time (30 min). This was attributed to the limited availability of uronic acid content of BSG. This was attributed to the BSG treatment by CP reducing the peak intensity at around 1596 cm^−1^ in a non-linear manner (Figure 2B). A similar result was reported by [11,28] for CP-treated gum Arabic. OH stretching vibration at 3500–3000 cm^−1^ indicates intra- and inter-molecular hydrogen bonds [29]. In addition, the peaks at 2921–2905 cm^−1^ were attributed to -CH_2_- and >CH- stretching and bending vibrations [29]. The samples treated for 20 and 30 min revealed a characteristic peak at 1354 cm^−1^. The structural modifications were more obvious in the sample CP_30_. Obviously, the intensity of the characteristic peak at 1718 cm^−1^ increased after plasma treatment, particularly at a longer time, indicating the weakness of some hydrogen bonds due to the presence of more hydrophobic regions in the BSG matrix. Moreover, the effect of CP treatment on the formation of carbonyl groups due to the oxidation of aldehydes, and hence the generation of di-aldehyde, increased this characteristic peak. Based on the report of Li et al. [30], gamma irradiation of xanthan gum enhanced the number of functional groups observed at 1730 cm^−1^. Gum Arabic treated with plasma also showed a new peak at 1722 cm^−1^, which was attributed to the formation of carbonyl groups [11].

#### 3.1.3. Contact Angle

Figure 3 reports the water contact angle (θ) of various BSG pellets. The lowest value of θ (56.23°) was measured in untreated BSG. In this sample, the deposited water droplet was immediately absorbed into the pellet. CP treatment reduced the hydrophilicity of BSG and higher θ values were measured by increasing the treatment time (58.96°, 71.43°, and 75.13° for CP_10_, CP_20_, and CP_30_, respectively). The rise of hydrophobicity was more obvious after CP treatment for 20 min. CP treatment induces changes in the chemical structure through different reactions and the breaking of chemical bonds that reduce the wettability of samples [31,32]. Topography is another factor influencing the contact angle. CP-induced etching increases the surface roughness (Section 3.1) and thus the surface hydrophobicity [31]. The changes in the density of polar (such as carboxyl and hydroxyl) groups and the surface topography of BSG after CP treatment were responsible for increasing surface hydrophobicity. The increase in the hydrophobicity was in accordance with the result of surface tension. Segat et al. [33] reported a higher hydrophobicity for whey protein isolate after plasma treatment. An increase in the hydrophobicity of gum Arabic (i.e., a significant rise of θ) was similarly reported by Amirabadi, Milani, and Sohbatzadeh [11] after plasma treatment for 20 min, mainly due to the changes in the surface anchors and chemical reactions within gum Arabic structure.

#### 3.1.4. Color Parameters

As reported in Table 1, the color attributes of various BSG samples remained unchanged after CP treatment. The *L**, *a**, *b**, and ΔE values of samples were in the range of 37.33–38.33, 11.33–12.00, 25.33–26.33, and 62.26–62.56, respectively. This observation showed that CP treatment did not affect the pigment and different colorant ingredients in BSG.

#### 3.1.5. Rheological Properties

##### Apparent Viscosity

Control and CP-treated BSG dispersions showed shear-thinning behavior (Figure 4). Similarly, Hosseini-Parvar et al. [34] reported the shear-thinning behavior for untreated BSG. This behavior is related to the disentanglement of the biopolymer network by increasing the shear rate and biopolymer chain alignment in the shear direction [35]. The higher viscosity at lower shear rates results in an appropriate mouthfeel during mastication [36], while lower viscosity at higher shear rates facilitates pumping and filling. At the shear rate of 15.7 s^−1^ and temperature of 20 °C, the apparent viscosity values of C, CP_10_, CP_20_, and CP_30_ dispersions were 105.7, 112.9, 138.4, and 144.8 mPa.s, respectively. CP treatment increased the apparent viscosity of BSG dispersion, especially at lower shear rate values that could be attributed to the chemical and structural changes in BSG (e.g., cross-linking through the formation of ester bonds). Cross-linking also increases the molecular weight and thus the excluded volume of BSG molecules. Zhang et al. [37] reported that polymerization, cross-linking, and other types of molecular interactions are responsible for increasing the viscosity at low shear rates after treatment with plasma. It was also reported that CP treatment develops more hydrophobic linkages and stronger intermolecular bonds in the treated gum dispersions [11]. Bulbul, Bhushette, Zambare, Deshmukh, and Annapure [13] studied the impact of CP treatment on xanthan gum. An increase in viscosity was reported by increasing power and duration of CP, which was attributed to cross-linking phenomenon. In another study on xanthan gum conducted by Zou et al. [38], the results of ATR-IR showed a decrease in the hydrogen bonding, which was ascribed to the xanthan gum cross-linking.

Table 2 represents the constant parameters of applied rheological models. Considering the values of the determination coefficient (R^2^), the rheological data of both native and CP-treated BSG dispersions were best fitted with the Casson model. The shear-thinning characteristic was confirmed by the values of the flow behavior index (n < 1) of the Power Law model. The increase in n values after CP treatment indicated a decrease in the pseudoplasticity of gum dispersions. A higher interaction between BSG chains after modification relatively enhanced the yield stress (τ_0_). Naji-Tabasi and Razavi [39] reported n, k, and τ_0_ values in the range of 0.36–0.48, 1.83–11.02 Pa.s^n^, and 0.8–3.85 Pa for different fractions of 1% BSG dispersion at 20 °C, respectively. Hosseini-Parvar, Matia-Merino, Goh, Razavi, and Mortazavi [34] reported dynamic τ_0_ values of 3.47 and 11.94 Pa for 1% and 2% BSG dispersions at 20 °C, respectively.

##### Effect of Temperature on Viscosity

Figure 5 shows the effect of the heating–cooling cycle on the viscosity of BSG dispersions. CP treatment significantly increased the viscosity of BSG dispersions over the studied temperature range. Therefore, the treatment of BSG by CP enhances the final viscosity even at high temperatures. The high viscosity of CP-modified BSG dispersions during heat treatment is desirable for practical food applications in which the hydrocolloids are used as thickeners. Regardless of the BSG type, a decrease in viscosity was observed during heating, which was related to the decrease in friction forces by increasing temperature. In other words, heating increases molecular movement as well as the breakdown of interactions [40]. A similar trend in apparent viscosity with an increase in temperature from 5 to 60 °C was previously reported by Razavi and Naji-Tabasi [41] for 1% BSG dispersion. At the same temperature, the control sample had a lower (28.5 to 20 °C) or equal (60 to 28.5 °C) viscosity during the cooling step as compared to the heating step. However, the CP-treated samples exhibited higher viscosity during cooling. The hysteresis, formed during the heating and cooling stages, was more obvious in the samples treated by CP at longer times (e.g., 20 and 30 min). This observation suggested the structural changes of BSG after modification by CP. In other words, CP treatment and subsequent heating improved the recovery rate of BSG during the cooling step likely due to the changes in the involved interactions (e.g., hydrophobic ones) and the formation of new structures.

##### Time Dependency

As depicted in Figure 6, samples C and CP_10_ exhibited relatively time-independent behavior at 50 s^−1^ during 900 s. However, CP_20_ and CP_30_ showed a slight time-dependent (i.e., rheopectic) character, likely as a result of increasing (hydrophobic) interactions among the BSG chains over time. The increase in the viscosity of CP_20_ and CP_30_ dispersions might be related to the higher recovery rate than the breakdown rate of intermolecular bonds at a constant temperature after a given time. The rheopexy could also be ascribed to the deformation and reorganization of the BSG network which increases the intermolecular associations [42]. For native BSG, Hosseini-Parvar [43] reported slight time-dependent (thixotropic) characteristics at a low concentration which was more obvious at higher concentrations due to the stronger interactions between the BSG chains.

##### Dynamic Rheological Properties

In gel systems, the loss modulus (G″) is smaller than the storage modulus (G′), while in dilute polysaccharide dispersions, G″ is always greater than G′. In concentrated polysaccharide dispersions and at low frequencies, G′ is dominant over G″, while at medium frequencies, these two moduli approach each other (i.e., cross-over occurs) [44,45]. As illustrated in Figure 7a,b, the G′ and G″ values of control and CP-modified BSG dispersions showed relatively similar dependency on frequency. BSG dispersions presented dominant elastic behavior (G′ > G′′) with no crossover (except for CP_10_) over the studied frequency range. Plasma treatment, particularly at longer treatment times, increased G′ values. The increase in viscoelastic parameters (G′ and G″) with the rise of treatment time was attributed to the formation of higher molecular interactions introduced in the FTIR section. There was a good agreement between the results of steady shear and dynamic tests. Amirabadi, Milani, and Sohbatzadeh [40] reported a similar effect of CP treatment on the rheological properties of gum Arabic. The loss factor results (or tan δ as the ratio of G″ to G′) are represented in Figure 7c. The values between 0.1–1 indicate the presence of weak gel structures [45]. Hosseini-Parvar [43] and Rafe et al. [46] similarly reported a solid-like behavior for native BSG dispersions. According to Figure 7d, complex viscosity (η*) was increased after CP modification, suggesting stronger and higher-ordered structures. In all samples, increasing angular frequency from 0.01–3.36 Hz reduced η* values that were attributed to the disruption of entanglements. This result is in good agreement with the results of shear-thinning properties. The consequence of network recovery and disruption might lead to various behaviors in the complex viscosity at the higher angular frequency (>3.6 Hz). The reduction of complex viscosity of native BSG in the range of 0.01 to 10 Hz was observed by Rafe and Razavi [47]. A similar behavior was also reported by Wei et al. [48] for fenugreek gum.

#### 3.1.6. Surface Tension

The surface tension of various BSG dispersions is reported in Figure 8. A significant reduction in the surface tension of water (72.1 ± 0.5 mN/m) was observed in the presence of BSG. Osano, Hosseini-Parvar, Matia-Merino, and Golding [5] similarly reported a decrease in the surface tension in the presence of various types of BSG. Both crude and protein-depleted BSGs were able to reduce the surface tension; however, the crude BSG was more effective, confirming the role of protein moiety in the adsorption to the interface and surface tension reduction [28]. The random coil structure, presence of some hydrophobic groups along the BSG backbone, and proteinaceous portion associated with the gum structure are the reasons for the surface activity of BSG [5]. There was a positive correlation between the CP duration and the ability of the gum sample to decrease the surface tension, indicating that the CP treatment alters the hydrophilic-hydrophobic character of gum. Misra, Yong, Phalak, and Jo [12] similarly reported that treating with plasma significantly reduced the interfacial tension of xanthan gum. Amirabadi, Milani, and Sohbatzadeh [11] reported that CP treatment for 20 min reduced the surface tension of gum Arabic from 45.95 to 44.24 mN/m. Surface hydrophobicity and chemical structure are the factors affecting surface tension [49]. Plasma treatment changes the functional groups on the surface, and thus affects the surface energy and activity [50].

### 3.2. Film Properties

The effects of incorporating different types of free and encapsulated forms of tannic acid and vitamin D_3_ on the visual appearance of modified BSG films are shown in Figure 9. The control film (without tannic acid and vitamin D_3_) was visually transparent and homogeneous with a smooth surface without any visible pores and cracks. The incorporation of tannic acid and vitamin D_3_, either in the free or encapsulated in form of nanophytosomes (92.20 ± 3.11 nm), showed a lower transparency and surface homogeneity than the control film. However, the free form of bioactive compounds resulted in a more noticeable surface roughness, which might be due to the relative aggregation of vitamin D_3_ in the casted film forming solution during drying. Similar results were also reported by other researchers in various biopolymer-based edible films incorporated with nanoparticles of hydrophobic bioactive compounds [19,23].

According to Table 3, the film thickness did not significantly change in the presence of free and encapsulated forms of tannic acid and vitamin D_3_, although the film thickness was numerically higher for BSG-free and BSG-phytospme compared to the control. This could be related to the relative instability of vitamin D_3_ droplets in the free form as well as the presence of phosphatidylcholine in the film forming solution of BSG-phytosome. In contrast to our results, an increase in the thickness of quince seed gum-based film after incorporation of vitamin D_3_ using an emulsion system was reported by Behjati and Yazdanpanah [23]. The film density significantly decreased with the addition of tannic acid and vitamin D_3_ (Table 3), which was more pronounced when added in their free forms. This was attributed to the relative surface inhomogeneity and higher thickness of BSG film after the addition of tannic acid and vitamin D_3_. As can be seen in Table 3, the incorporation of tannic acid and vitamin D_3_ could increase the opacity of BSG films due to the light scattering effect of the dispersed phytospmes and free form of vitamin D_3_ droplets [51]. Additionally, moisture content and solubility were significantly affected by the incorporation of phytosomes and vitamin D_3_ droplets. The highest moisture content (9.91%) and solubility (48.13%) were obtained for the control film. The significant (*p* < 0.05) decreases in the water solubility and moisture content of BSG-phytosome and BAS-Free films could be related to their higher hydrophobic nature.

Table 4 shows the color parameters of the modified BSG films. Incorporating tannic acid and vitamin D_3_ decreased (*p* < 0.05) *L** and WI values compared to the control BSG film which were more noticeable in the presence of phytosome droplets. The *a**, *b**, and ΔE values increased significantly in BSG-phytosome and BSG-Free samples, suggesting these films tended toward yellowness and redness.

Figure 10 displays the reducing trend of both tannic acid and vitamin D_3_ concentrations in the BSG-phytosome and BSG-free films during storage. However, the encapsulation of tannic acid and vitamin D_3_ in the nanophytosome structure resulted in pronounced higher stability than free forms. High stability of tannic acid with antioxidant, antimicrobial, and astringent potential after encapsulation using solid nanoparticles was reported previously [52]. Additionally, Behjati and Yazdanpanah [23] observed the reduction of vitamin D_3_ in the quince seed gum film using nanoemulsion and emulsion loading techniques.

## 4. Conclusions

The influence of CP treatment on the various properties of BSG was investigated. Plasma treatment increased the surface roughness of the samples. The FT-IR spectra showed some changes in the functional groups of BSG, which altered the balance between polar and nonpolar groups, and as a result, changed the surface tension and water contact angle. The modified BSG samples showed a greater contact angle than the unmodified sample due to the changes in the gum hydrophobicity. BSG modification improved the ability of gum to decrease the surface tension. Both unmodified and CP-modified gums had shear thinning properties. Plasma treatment affected temperature- and time-dependent rheological characteristics. Treated samples revealed higher apparent viscosity and improved viscoelastic properties than the control sample. In conclusion, plasma treatment can be considered a powerful non-thermal tool to modify the rheological, structural, and surface properties of BSG for further applications. The modified BSG gum was successfully used to form good functional edible films. The addition of tannic acid and vitamin D_3_ into the film-forming dispersion of BSG using the nanophytosome structure resulted in physical properties changes of the final films. The stability of tannic acid and vitamin D_3_ encapsulated in the phytosome form was higher than the addition of the free bioactive compounds to the modified BSG film during 14 days of storage.

## Figures and Tables

**Figure 1 foods-12-00071-f001:**
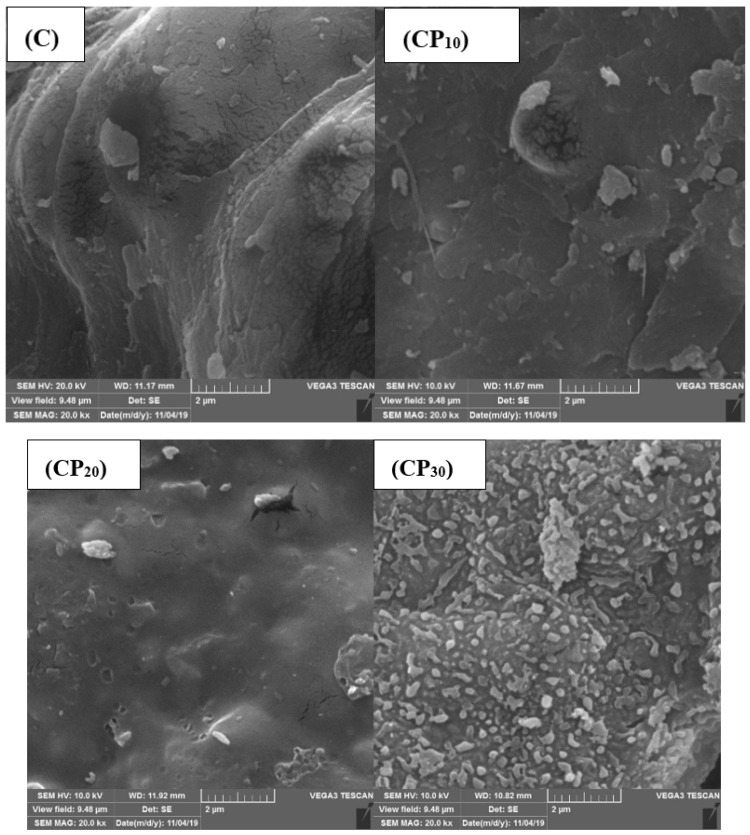
SEM micrographs (×20,000 magnification) of BSG powder treated by cold plasma (CP); C, CP_10_, CP_20_, and CP_30_ indicate unmodified samples and those treated by CP for 10, 20, and 30 min, respectively.

**Figure 2 foods-12-00071-f002:**
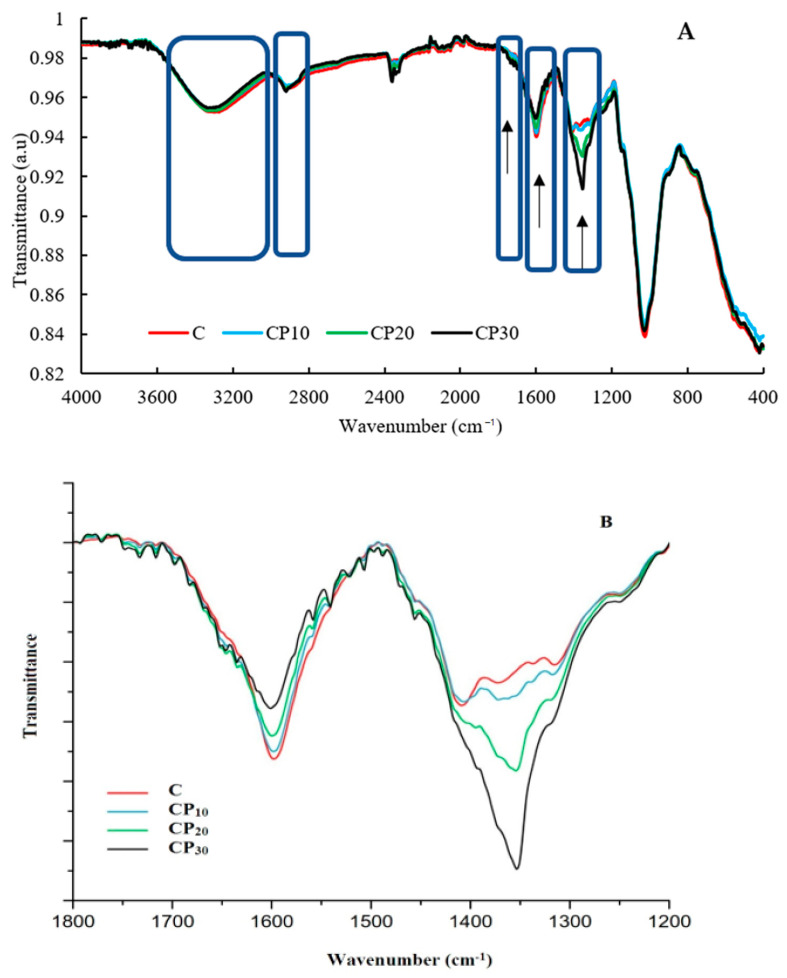
FT-IR spectra of different BSG samples at 4000–400 cm^−1^ (**A**) and 1800–1200 cm^−1^ (**B**); C, CP_10_, CP_20_, and CP_30_ indicate unmodified samples and those treated by CP for 10, 20, and 30 min, respectively.

**Figure 3 foods-12-00071-f003:**
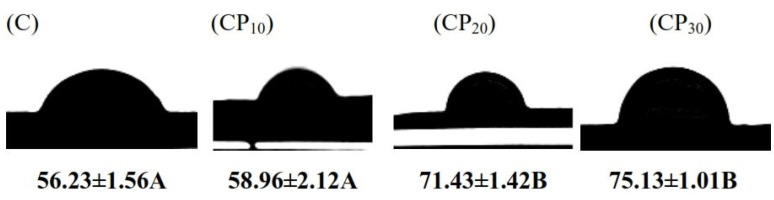
Water contact angle of various BSG samples; C, CP_10_, CP_20_, and CP_30_ indicate unmodified samples and those treated by CP for 10, 20, and 30 min, respectively. Different uppercase letters indicate significant differences (*p* < 0.05).

**Figure 4 foods-12-00071-f004:**
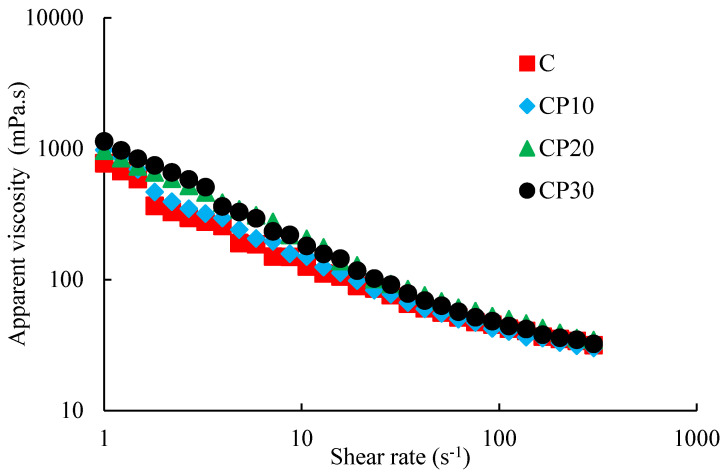
Changes in the apparent viscosity of different BSG dispersions (0.3%) at 20 °C as a function of shear rate; C, CP_10_, CP_20_, and CP_30_ indicate unmodified samples and those treated by CP for 10, 20, and 30 min, respectively.

**Figure 5 foods-12-00071-f005:**
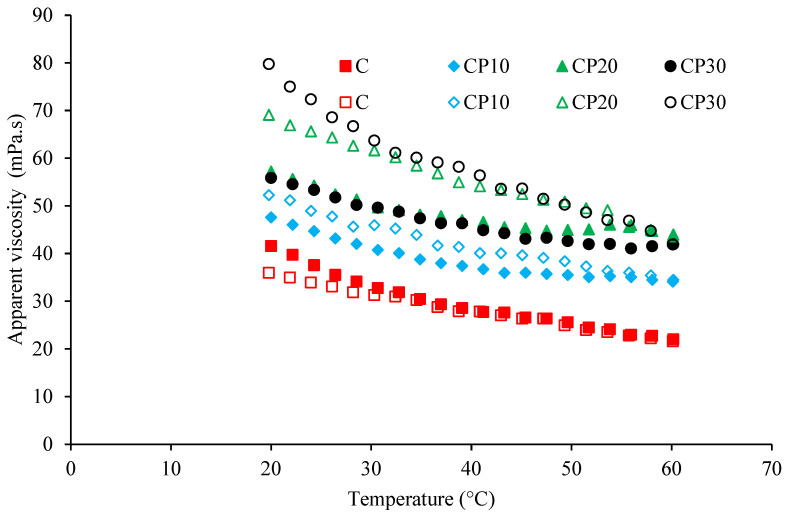
Changes in the apparent viscosity of different BSG dispersions (0.3%) at a constant shear rate of 50 s^−1^ as a function of temperature; C, CP_10_, CP_20_, and CP_30_ indicate unmodified samples and those treated by CP for 10, 20, and 30 min, respectively. Filled and open symbols represent heating and cooling steps, respectively.

**Figure 6 foods-12-00071-f006:**
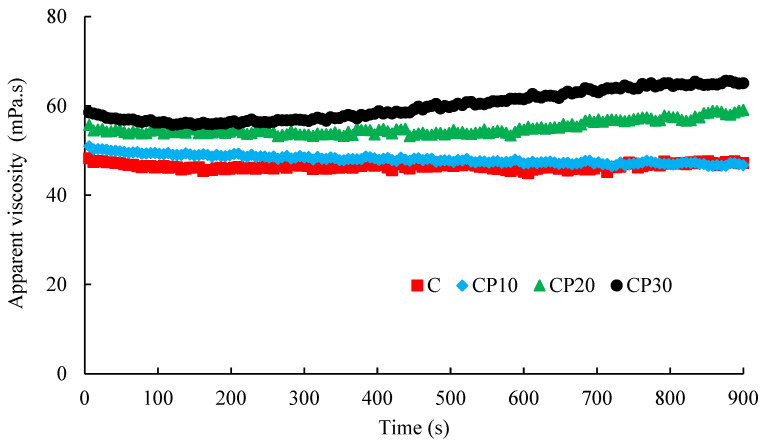
Changes in the apparent viscosity of different BSG dispersions (0.3%) at a constant shear rate of 50 s^−1^ as a function of time; C, CP_10_, CP_20_, and CP_30_ indicate unmodified samples and those treated by CP for 10, 20, and 30 min, respectively.

**Figure 7 foods-12-00071-f007:**
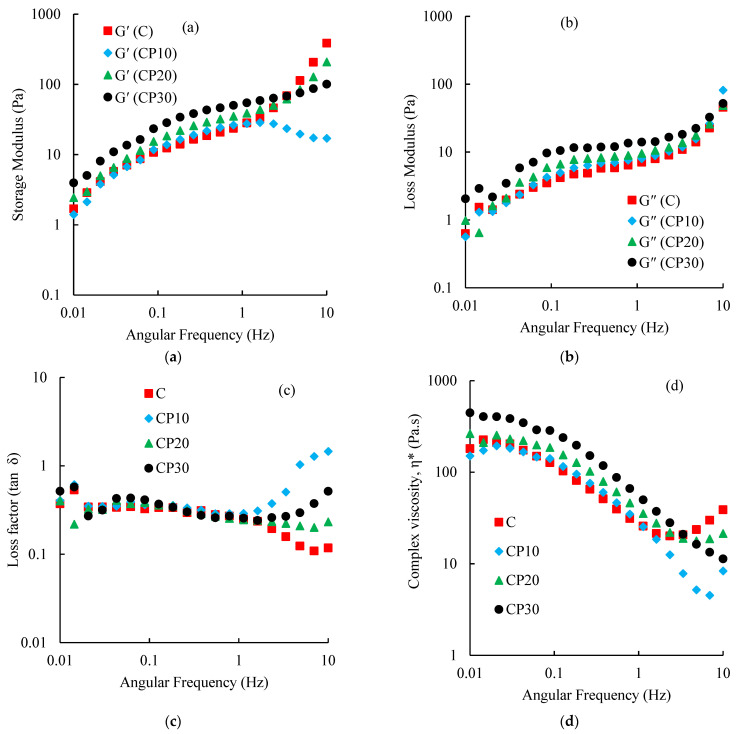
Changes in the storage modulus (**a**), loss modulus (**b**), loss factor (**c**), and complex viscosity (**d**) of different BSG dispersions (0.3%) at 20 °C as a function of angular frequency; C, CP_10_, CP_20_, and CP_30_ indicate unmodified samples and those treated by CP for 10, 20, and 30 min, respectively.

**Figure 8 foods-12-00071-f008:**
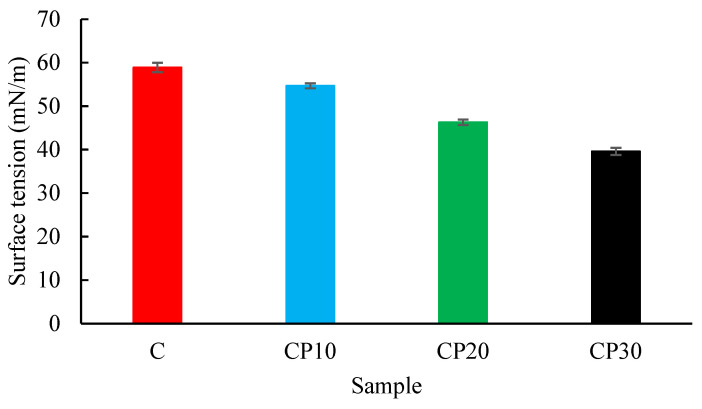
Changes in the Surface tension of different BSG dispersions (0.3%) at 20 °C; C, CP_10_, CP_20_, and CP_30_ indicate unmodified samples and those treated by CP for 10, 20, and 30 min, respectively.

**Figure 9 foods-12-00071-f009:**
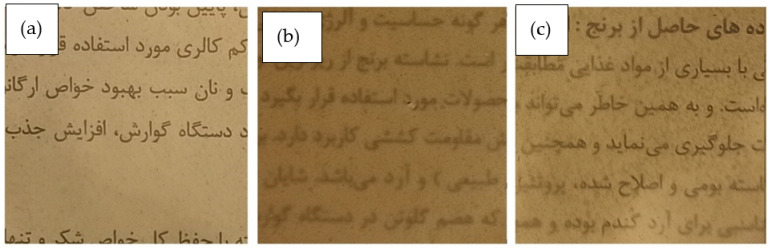
Appearance of (**a**) control modified BSG edible film and those incorporated with tannic acid and vitamin D_3_ in (**b**) encapsulated form in nanophytosome and (**c**) free form.

**Figure 10 foods-12-00071-f010:**
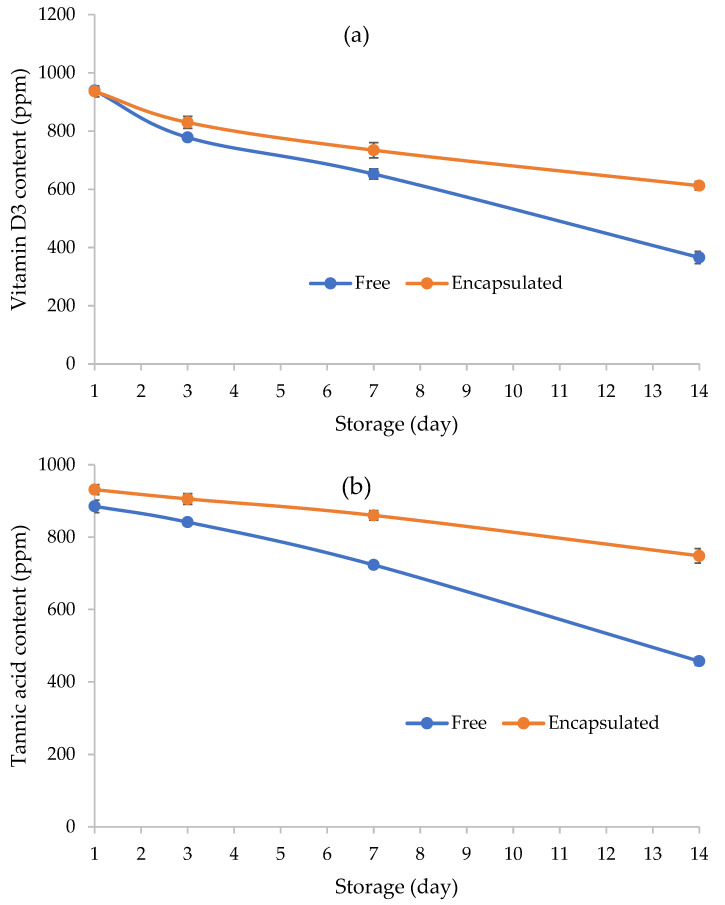
The stability of vitamin D_3_ (**a**) and tannic acid (**b**) in the modified BSG film in encapsulated form in nanophytosome and free form over storage time (day).

**Table 1 foods-12-00071-t001:** Color parameters of various BSG samples.

Sample	*L**	*a**	*b**	ΔE
C	37.67 ± 1.53 A	11.67 ± 0.58 A	25.67 ± 0.58 A	62.46 ± 1.33 A
CP_10_	37.33 ± 1.53 A	11.33 ± 0.58 A	25.33 ± 0.58 A	62.56 ± 1.50 A
CP_20_	37.33 ± 0.58 A	12.00 ± 1.00 A	25.33 ± 0.58 A	62.34 ± 1.27 A
CP_30_	38.33 ± 1.15 A	11.67 ± 0.58 A	26.33 ± 1.15 A	62.26 ± 1.21 A

Data represent the average of three independent replicates ± standard deviation. C, CP_10_, CP_20_, and CP_30_ indicate unmodified samples and those treated by CP for 10, 20, and 30 min, respectively. Significant differences were not observed in each column.

**Table 2 foods-12-00071-t002:** Apparent viscosity and rheological parameters of different BSG dispersions (0.3%) at 20 °C.

Sample		C	CP_10_	CP_20_	CP_30_
Apparent viscosity (mPa.s)		105.72	112.94	138.35	144.84
Rheological model					
Power Law	k (Pa.s^n^)	0.46	0.39	0.38	0.35
	n	0.53	0.68	0.89	0.96
	R^2^ (%)	95.54	94.14	96.79	93.92
Casson	k_c_ (Pa.s^0.5^)	0.14	0.12	0.13	0.12
	τ_0_ (Pa)	0.84	0.90	0.98	0.99
	R^2^ (%)	99.76	99.66	99.33	99.53
Bingham	μ (Pa.s)	0.030	0.027	0.031	0.029
	τ_0_ (Pa)	1.02	1.17	1.59	1.56
	R^2^ (%)	98.79	98.99	97.99	99.06
Herschel–Bulkely	k (Pa.s^n^)	0.64	0.82	1.06	1.30
	n	0.41	0.34	0.34	0.27
	τ_0_ (Pa)	0.56	0.67	0.67	0.93
	R^2^ (%)	92.70	89.98	94.68	86.14

C, CP_10_, CP_20_, and CP_30_ indicate unmodified samples and those treated by CP for 10, 20, and 30 min, respectively.

**Table 3 foods-12-00071-t003:** Physical properties of control modified BSG edible film and those incorporated with tannic acid and vitamin D_3_ in the encapsulated form in nanophytosome and free form.

	BSG-C	BSG-Phytosome	BSG-Free
Thickness (μm)	55.03 ± 7.33 A	63.50 ± 12.70 A	67.73 ± 14.66 A
Density (g/cm^3^)	1.15 ± 0.02 A	1.03 ± 0.01 B	0.94 ± 0.02 C
Opacity (1/mm)	7.67 ± 0.19 C	8.58 ± 0.10 B	9.54 ± 0.12 A
Moisture content (%)	9.91 ± 0.47 A	7.36 ± 0.46 B	8.34 ± 0.76 B
Solubility (%)	48.13 ± 0.41 A	38.76 ± 1.26 C	42.75 ± 1.49 B

Data represent the average of three independent replicates ± standard deviation. In each row, different letters indicate significant differences (*p* < 0.05).

**Table 4 foods-12-00071-t004:** Color properties of control modified BSG edible film and those incorporated with tannic acid and vitamin D_3_ in the encapsulated form in nanophytosome and free form.

	BSG-C	BSG-Phytosome	BSG-Free
*L**	63.33 ± 1.15 A	48.00 ± 1.00 C	52.33 ± 1.15 B
*a**	5.33 ± 0.58 C	9.00 ± 1.00 A	7.33 ± 0.58 B
*b**	30.33 ± 1.15 C	34.33 ± 0.58 A	32.67 ± 0.58 B
ΔE	43.03 ± 0.06 C	57.62 ± 0.90 A	52.98 ± 0.63 B
WI	52.09 ± 0.15 A	37.03 ± 0.92 C	41.74 ± 0.70 B

Data represent the average of three independent replicates ± standard deviation. In each row, different letters indicate significant differences (*p* < 0.05).

## Data Availability

The data used to support the study are included within the article. More information can be obtained by contacting the corresponding author.

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
