# Peer review of "Atmospheric Pressure Cold Plasma Modification of Basil Seed Gum for Fabrication of Edible Film Incorporated with Nanophytosomes of Vitamin D3 and Tannic Acid"

_foods, 2022, doi:10.3390/foods12010071_

Round 1

Reviewer 1 Report

Dear Authors,

The manuscript is design and presented in well manner therefore, i suggest some corrections before further processing of the manuscript.

The results of best treatments should be mention clearly in abstract section

Is cold plasma treatment suitable to reducing hygroscopic nature of BSG?

Material and methods are ok

Why CP 30 peak shifted higher in FTIR?

How CP treatment attributed to formation of carbonyl group?

Why the color scale value of L* and b* increased in CP treated samples, whereas control was reported higher than CP 10 and CP 20.

Results of color properties should be describe briefly.

Author Response

Reviewer #1

The manuscript is design and presented in well manner therefore, i suggest some corrections before further processing of the manuscript.

 Response: We appreciate your positive outlook and constructive comments.

The results of best treatments should be mention clearly in abstract section

Response: More details were added to the revised manuscript based on your comment. Please check page 1.

Is cold plasma treatment suitable to reducing hygroscopic nature of BSG?

Response: As discussed in section 3.1.3, CP can affect the chemical structure of BSG through different reactions and the breaking of chemical bonds that reduce the wettability of samples. Topography was also another factor influencing on contact angle. CP-induced etching increases the surface roughness and thus the surface hydrophobicity. Therefore, the induced changes in the density of polar (such as carboxyl and hydroxyl) groups and the surface topography of BSG after CP treatment were responsible for increasing surface hydrophobicity.

Material and methods are ok

Response: Thank you for your positive outlook.

Why CP 30 peak shifted higher in FTIR?

Response: According to Fig. 2, the location of appeared peaks did not show any significant changes. However, the intensity of peaks changed after modification which was attributed to the irradiation effects on the functional groups. More clear explaniations were added in the revised manuscript.

How CP treatment attributed to formation of carbonyl group?

Response:This was most likely due to the oxidation of aldehydes, and hence the generation of di-aldehyde. Ali, Ganie, and Mazumdar (2018) and Sarika and James (2015) also reported similar results in the preparation of oxidized gum Arabic.

Ali, A., Ganie, S. A., & Mazumdar, N. (2018). A new study of iodine complexes of oxidized gum Arabic: An interaction between iodine monochloride and aldehyde groups. Carbohydrate Polymers, 180, 337–347. https://doi.org/10.1016/j. carbpol.2017.10.005

Sarika, P., & James, N. R. (2015). Preparation and characterisation of gelatin–gum Arabic aldehyde nanogels via inverse miniemulsion technique. International Journal of Biological Macromolecules, 76, 181–187. https://doi.org/10.1016/j. ijbiomac.2015.02.038.

Why the color scale value of L* and b* increased in CP treated samples, whereas control was reported higher than CP 10 and CP 20.

Response: As reported in Table 1, plasma treatment at various times had no significant effects on the color properties of BSC samples.

Results of color properties should be describe briefly.

 Response: Based on your comment, the color properties were explained briefly in the revised manuscript. Please check page 7.

Reviewer 2 Report

Why are the films discussed? Why not using CP10 and CP20 for film production?

Abstract: “FT-IR spectra of CP-treated BSG revealed change at 1596 and 1718 cm^-1.” What does that mean? The intensity higher, lower, additional peaks? Which groups are responsible?

“… D-mannose, L-arabinose, L-rhamnose, and L-arabinose …” just one L-arabinose

The following papers must be discussed with respect to physicochemical and rheological properties of BSG:
- Radio-frequency (RF) room temperature plasma treatment of sweet basil seeds (Ocimum basilicum L.) for germination potential enhancement by immaculation
- Evaluation of Atmospheric Cold Plasma Technique Activity on Phenylpropanoids Gene Expression and Essential Oil Contents and Different Traits of Ocimum basilicum L.
- An application of cold atmospheric plasma to enhance physiological and biochemical traits of Basil
- Cold plasma treatment strategies for the control of fusarium oxysporum f. sp. basilici in sweet basil
- Radiofrequency cold plasma treatment enhances seed germination and seedling growth in variety CIM-Saumya of sweet basil (Ocimum basilicum L.)

Regarding these papers is it still correct: “To the best of our knowledge, the effect of CP treatment on the physicochemical and rheological properties of BSG has not yet been reported.”

Equation 4: 0.5 on the left hand side must be in the exponent.

Line 108: “-1” to exponent

Line 109: n and 0.5 in exponent. Take care Pa*(s^n) and (Pa*s)^0.5 or you have to write in equation 4 …+(kc*y)^0.5 (exponent before multiplication before addition)

Did you really use distilled water or was it deionized water?

Line 195: “-1” to exponent

Figure 2: Unit of transmittance is missing

Line 202 and 205: “-1” to exponent

Are there new peaks or just increase in intensity in Figure 2?

Table 1: Just A is presented, so it is better to mention no significant differences and not “In each column, different uppercase letters indicate significant differences (p<0.05).”

Figure 4: “a” is waste

“R2” better R^2

“best fitted with the Casson model” what was the goodness parameter? Just R^2 or sum of least squares?

Table 2, Casson parameter kc the c is missing. Take care about the unit (see comment of line 109)

Table 2: sometimes 2 significant digits sometimes just one; does this makes sense?

Figure 4b should be 5 and follows.

In the Figures, no error bars can be seen. Are these measurement just performed once?

Line 335: “Tan” better tan

“Osano, Hosseini-Parvar, Matia-Merino and Golding [5]” better Osano et al. [5]

“Misra, Yong, Phalak and Jo [12]” better Misra et al. [12]. Please check all references, there are more!

What is the meaning of the text shown in Figure 7. If text is shown, then everyone should be able to read it otherwise remove it.

Author Response

Reviewer # 2

Why are the films discussed? Why not using CP10 and CP20 for film production?

Response: This research was done in two separate parts. In the first part, the effects of cold plasma treatment on the physicochemical and structural properties of BSG dispersion were studied; whereas, in the second part, the best CP-modified BSG sample (CP30) was selected based on rheological measurements and surface characteristics to prepare functional films incorporated with free and encapsulated forms of vitamin D3 and Tannic acid.

Abstract: “FT-IR spectra of CP-treated BSG revealed change at 1596 and 1718 cm^-1.” What does that mean? The intensity higher, lower, additional peaks? Which groups are responsible?

Response: Many thanks for your comment. More detailed explanations were added to the FTIR results. Please check section 3.1.2.

“… D-mannose, L-arabinose, L-rhamnose, and L-arabinose …” just one L-arabinose

Response: It was corrected in the revised manuscript.

The following papers must be discussed with respect to physicochemical and rheological properties of BSG:
- Radio-frequency (RF) room temperature plasma treatment of sweet basil seeds (Ocimum basilicum L.) for germination potential enhancement by immaculation
- Evaluation of Atmospheric Cold Plasma Technique Activity on Phenylpropanoids Gene Expression and Essential Oil Contents and Different Traits of Ocimum basilicum L.
- An application of cold atmospheric plasma to enhance physiological and biochemical traits of Basil
- Cold plasma treatment strategies for the control of fusarium oxysporum f. sp. basilici in sweet basil
- Radiofrequency cold plasma treatment enhances seed germination and seedling growth in variety CIM-Saumya of sweet basil (Ocimum basilicum L.)

Regarding these papers is it still correct: “To the best of our knowledge, the effect of CP treatment on the physicochemical and rheological properties of BSG has not yet been reported.”

Response: Many thanks for your comment. The purpose of this work was to investigate the effect of cold plasma (CP) on the rheological, structural, and surface characteristics of Basil Seed Gum (BSG, as a hydrocolloid which is extracted from basil seeds) and not on the germination of basil seeds or the traits of Ocimum basilicum as a medicinal plant. To our knowledge, this is the first report on the effect of CP on BSG. After that, the possible application of CP-modified hydrocolloid was studied in the fabrication of edible films containing free and encapsulated forms of vitamin D3 and Tannic acid.

Equation 4: 0.5 on the left hand side must be in the exponent.

Response: Thank you very much. It was corrected.

Line 108: “-1” to exponent

Response: Thank you very much. It was corrected.

Line 109: n and 0.5 in exponent. Take care Pa*(s^n) and (Pa*s)^0.5 or you have to write in equation 4 …+(kc*y)^0.5 (exponent before multiplication before addition)

Response: Thank you for your valuable comments. The manuscript was corrected. In equations 1, 2, and 4, the exponent beloged to the shear rate (γ) and not (k*γ).

Did you really use distilled water or was it deionized water?

Response: Double distillated water was used in the preparation of all dispersions.

Line 195: “-1” to exponent

Response: Thank you for pointing out this issue. It was modified accordingly in the revised manuscript.

Figure 2: Unit of transmittance is missing

Response: The unit of transmittance (a.u) was added in Fig. 2 in the revised manuscript.

Line 202 and 205: “-1” to exponent

Response: Thank you very much. It was corrected.

Are there new peaks or just increase in intensity in Figure 2?

Response: According to Fig.2, the location of appeared peaks did not show any significant changes. However, the intensity of peaks changed after modification which was attributed to the irradiation effects on the functional groups. More clear explaniations were added to the revised manuscript.

Table 1: Just A is presented, so it is better to mention no significant differences and not “In each column, different uppercase letters indicate significant differences (p<0.05).”

 Response: Thank you very much for your helpful comment. The manuscript was modified.

Figure 4: “a” is waste

Response: Thank you very much for your helpful comment. The manuscript was modified.

“R2” better R^2

Response: Thank you. It was modified based on your comment in the revised manuscript.

“best fitted with the Casson model” what was the goodness parameter? Just R^2 or sum of least squares?

Response:  R^2 was used to select the best model.

Table 2, Casson parameter kc the c is missing. Take care about the unit (see comment of line 109)

Response: The manuscript was corrected.

Table 2: sometimes 2 significant digits sometimes just one; does this makes sense?

Response: Now, for a same attribute (parameter), the number of significant digits is as same as each other in Table 2.

Figure 4b should be 5 and follows.

Response: Thank you for careful evaluation. The manuscript was corrected.

In the Figures, no error bars can be seen. Are these measurement just performed once?

Response: Thank you for your question. As mentioned in section 2.7, all measurements were done in triplicate. Reporting standards errors or standard deviations are not routine for rheology or FT-IR data. However, for the other experiments, the error bars were added.

Line 335: “Tan” better tan

Response: It was corrected.

“Osano, Hosseini-Parvar, Matia-Merino and Golding [5]” better Osano et al. [5]

“Misra, Yong, Phalak and Jo [12]” better Misra et al. [12]. Please check all references, there are more!

Response: This format of citation is based the author guide in this journal (MDPI endnote format on the website of Foods).

What is the meaning of the text shown in Figure 7. If text is shown, then everyone should be able to read it otherwise remove it.

Response: The text beneath the films (in Persian fonts) does not have any meaning. This figure (now Figure 9) only shows the visual appearance, color and appropriate transparency of various edible films for future applications.